# Long-Term Outcomes of Intravesical Mitomycin C Administered via Electromotive Drug Administration or Conductive Chemo-Hyperthermia in Non-Muscle-Invasive Bladder Cancer

**DOI:** 10.3390/cancers17030453

**Published:** 2025-01-28

**Authors:** Maria Teresa Melgarejo-Segura, Alberto Zambudio-Munuera, Miguel Ángel Arrabal-Polo, Pablo Lardelli-Claret, Manuel Pareja-Vilchez, Miguel Arrabal-Martín

**Affiliations:** 1Department of Urology, San Cecilio Clinical University Hospital of Granada, Av. del Conocimiento, s/n, 18007 Granada, Spain; 2ibs.GRANADA Biosanitary Research Institute, Av. de Madrid, 15, Beiro, 18012 Granada, Spain; 3Faculty of Medicine, University of Granada, Avenida Doctor Jesús Candel Fábregas No. 11, 18071 Granada, Spain

**Keywords:** bladder cancer, non-muscle-invasive bladder cancer, HIVEC, EMDA, mitomycin C, recurrence prevention, chemotherapy

## Abstract

Bladder cancer is one of the most common types of cancer worldwide, and many cases are classified as non-muscle-invasive, meaning they are contained within the bladder but can still return or progress over time. Traditional treatments for these cases, like Bacillus Calmette–Guérin (BCG) therapy, can be effective but often have limitations, such as side effects and recurrence. This research investigates two promising alternative treatments—hyperthermia-induced potentiation of mitomycin C (HIVEC) and electromotive drug administration (EMDA)—to see which offers better long-term outcomes for patients. This study compares the effectiveness of these two therapies in preventing cancer recurrence and progression in patients at intermediate and high risk. By providing long-term data, the findings could help doctors choose the most suitable treatment for patients and guide future research in bladder cancer therapies.

## 1. Introduction

Bladder cancer (BC) is the 10th most common cancer worldwide, with nearly 600,000 new cases diagnosed annually, and is one of the most costly malignancies in terms of diagnosis, treatment, and follow-up [1]. Approximately 75% of these cases are classified as non-muscle-invasive bladder cancer (NMIBC) after transurethral resection of the bladder tumor (TURBT). Nearly half of these NMIBCs are at intermediate or high risk for recurrence and progression [2], highlighting the need for effective adjuvant intravesical therapies. Bacillus Calmette–Guérin (BCG) is the gold-standard treatment for intermediate- and high-risk NMIBC, but recurrence remains a concern [3]. Moreover, BCG is associated with systemic adverse events [4] and has faced supply shortages [5]. Therefore, alternative strategies are needed to reduce recurrence in intermediate-risk patients and improve survival in high-risk cases.

The exploration of non-BCG adjuvant treatments has gained momentum. Two promising strategies for improving the efficacy of intravesical therapy are the use of hyperthermia-induced potentiation of mitomycin C (HIVEC) and electromotive drug administration (EMDA) to enhance drug delivery into the bladder wall. Both approaches leverage technological advances to optimize the pharmacokinetics of mitomycin C, a widely utilized chemotherapeutic agent for NMIBC. While studies have demonstrated the potential of each modality individually, few comparisons exist, especially with extended follow-up.

This study builds on previously published short-term results, extending the follow-up period to evaluate the long-term efficacy and outcomes of these two innovative modalities [6].

The objective of this study is to evaluate and compare the efficacy of intravesical mitomycin C administration enhanced by conductive hyperthermia (HIVEC) versus mitomycin C administration via electromotive drug administration (EMDA) in patients with intermediate- and high-risk non-muscle-invasive bladder cancer (NMIBC), using the cohort with the longest follow-up reported to date comparing these two devices. Additional objectives included assessing the influence of individual risk factors on treatment outcomes.

## 2. Materials and Methods

### 2.1. Design

A prospective observational study was conducted at a Spanish center, including patients diagnosed with intermediate- and high-risk non-muscle-invasive bladder cancer (NMIBC) without carcinoma in situ (CIS). This updated analysis expands upon a previously published study [1], incorporating patients treated between August 2018 and December 2024.

This study adhered to the ethical standards of the Declaration of Helsinki and obtained written informed consent from all participants. This study was approved by the local ethics committee before it began.

### 2.2. Inclusion Criteria

Patients diagnosed with intermediate- or high-risk NMIBC, either primary or recurrent, were eligible for this study. Recurrent patients could participate if they did not receive treatment with EMDA or HIVEC within the two years prior to enrollment.

### 2.3. Exclusion Criteria

Exclusion criteria included a history of previous or concomitant carcinoma in situ (CIS), allergy to mitomycin C (MMC), non-urothelial bladder carcinomas, or upper urinary tract urothelial carcinoma at the time of diagnosis. Other exclusion factors were inadequate bone marrow reserves (white blood cell count < 3000 × 10^6^ cells per liter; platelet count < 100 × 10^9^ per liter), liver or kidney function levels exceeding twice the normal laboratory reference values, untreated urinary tract infections, bladder capacity less than 150 mL, chemotherapy within the past 3 months, prior pelvic radiotherapy, and pregnancy. Additionally, patients with a history of T2 tumors, non-urothelial carcinomas, an ECOG performance status greater than 2, or breastfeeding women were excluded.

### 2.4. Interventions

The initial evaluation included urinary cytology, cystoscopy, and upper urinary tract imaging. Visible bladder tumors were resected, and random biopsies were taken if cytology was positive and a non-papillary tumor was present. A second transurethral resection of bladder tumor (TURBT) was performed within 2–4 weeks if resection was incomplete or muscle was absent in the sample. Upper urinary tract pathology was excluded via computed tomography urography (CTU), and urinary tract infection (UTI) was confirmed to be negative by urine culture. Instillations began 4–6 weeks post-TURBT.

The decision regarding the treatment allocation was made based on a thorough discussion between the patient and the medical team, who provided detailed information about the available treatment options, the supporting evidence, and the potential adverse effects. As both therapies were accessible concurrently, patients had the opportunity to choose the treatment that best suited their preferences and individual circumstances. We also provided standard care treatment in accordance with the guidelines of the European Association of Urology (EAU) [7].

Conductive chemo-hyperthermia was administered using the Combat BRS System V2.0 (Combat Medical Ltd., Wheathampstead, UK), a device designed to heat 40 mg of mitomycin C (MMC) diluted in 50 mL of distilled water to 43 ± 0.5 °C. The heated solution was then instilled into the bladder at a flow rate of 200 mL/min for 60 min.

Electromotive force-assisted chemotherapy was performed using the EMDA device (Physionizer^®^ 30, Physion^®^, Medolla, Italy). This device applies an electrical current through a 16 Fr catheter, which acts as the anode, delivering 40 mg of MMC (diluted in 50 mL of distilled water) into the bladder. A cathode patch is placed on the hypogastrium of the patient. The treatment was administered at a current of 20 mA for 30 min.

Both treatments follow the same protocol: an initial induction phase with six weekly instillations, followed by six monthly maintenance sessions.

### 2.5. Follow-Up

Patient follow-up was conducted every three months, with evaluations including cystoscopy, urinary cytology, and biopsy of all visible tumors, as well as healthy bladder mucosa in cases of positive cytology. A radiological study of the upper urinary tract, either through ultrasound or computed tomography urography (CTU), was performed every six months. If cystoscopy showed abnormalities or if imaging tests raised suspicion of recurrence, a transurethral resection (TUR) was performed. If cytology was positive, biopsies of the bladder mucosa were taken.

Patients without evidence of progression or recurrence at the last follow-up visit were censored. Those lost to follow-up were censored based on the last known date of survival. Recurrence was defined by the reappearance of the tumor unless muscle-invasive cancer, metastases, or local disease progression (T3–T4) were detected on radiological imaging, in which case progression was noted.

### 2.6. Statistical Analysis

Data were collected prospectively. Categorical and continuous variables were analyzed using chi-square tests, Fisher’s exact test, and Student’s *t*-test. Time-to-event outcomes were evaluated with the Kaplan–Meier method, and survival curves were generated for each study arm. Group comparisons were conducted using the log-rank test. All analyses were two-sided, with *p*-values < 0.05 considered statistically significant. Statistical analyses were performed using SPSS version 23.0 (Armonk, NY, USA: IBM Corp.).

In addition to the primary analyses, exploratory evaluations were performed using stratification to assess consistency of efficacy between the treatment arms. Hazard ratios (HRs) and 95% confidence intervals (CIs) for these endpoints were estimated using Cox proportional hazards models. *p*-values for these analyses were derived from the log-rank test.

## 3. Results

### 3.1. Patients

Between August 2018 and December 2024, the cohort from the previously published study, which initially included 56 patients in the HIVEC group and 42 patients in the EMDA group [7], was expanded by the inclusion of 11 new patients in the HIVEC group and 15 in the EMDA group. This resulted in 67 patients in the HIVEC group and 57 in the EMDA group. However, during the follow-up period, two patients in the EMDA group relocated and were unable to continue in this study, and one withdrew consent to participate. Consequently, the final sample sizes were 67 patients in the HIVEC group and 54 patients in the EMDA group (Figure 1).

Patient demographics and tumor characteristics are summarized in Table 1. Both groups were comparable in terms of baseline characteristics, with no statistically significant differences observed between them (all *p* > 0.05). Most patients in both groups were male (HIVEC: 89.6%, EMDA: 83.3%) and aged 70 years or younger (HIVEC: 64.2%, EMDA: 63.0%). A majority of the cases were recurrent tumors (HIVEC: 76.1%, EMDA: 77.8%), with tumors frequently presenting as multiple (HIVEC: 79.1%, EMDA: 83.3%) and larger than 3 cm (HIVEC: 58.2%, EMDA: 63.0%). In terms of tumor stage, the distribution of Ta and T1 tumors was similar between the groups. Most tumors were classified as high grade according to the 2004/2016 WHO grading system (HIVEC: 88.1%, EMDA: 90.7%) and were categorized as high risk based on the EAU classification (HIVEC: 94.0%, EMDA: 92.6%).

### 3.2. Efficacy

The median follow-up period was 38.6 months (interquartile range: 23.8–53.4 months)). At the 36-month follow-up, the disease-free survival (DFS) rate was 62.4% (95% CI: 49–73%) in the HIVEC group and 67% (95% CI: 52–78%) in the EMDA group. The adjusted hazard ratio (HR) for the treatment effect, with HIVEC as the reference group, was 0.95 (95% CI: 0.54–1) (long-rank *p* = 0.4; Figure 2, Table 2).

At the end of follow-up, only one patient in the EMDA group progressed to T2 disease.

Exploratory analyses with stratification by NMIBC risk factors were performed. The results presented in the forest plot show that none of the evaluated risk factors, including tumor size, stage, grade, and multiplicity, demonstrated a significant impact on treatment outcomes. Hazard ratios for tumor size ≤3 cm versus >3 cm were 0.92 and 1.16, for Ta versus T1 stage were 0.59 and 1.63, and for low grade versus high grade were 0.32 and 1.17. Additionally, no significant differences were observed between patients with single versus multiple tumors (HR: 0.84 and 1.04) or between primary versus recurrent tumors (HR: 1.07) (Figure 3).

## 4. Discussion

Our study compared the efficacy of intravesical mitomycin C administration enhanced by conductive hyperthermia (HIVEC) versus electromotive drug administration (EMDA) in patients with intermediate- and high-risk non-muscle-invasive bladder cancer (NMIBC) within the context of the longest follow-up cohort reported to date. The results showed no significant differences in disease-free survival (DFS) rates at 36 months between the two groups (62.4% for HIVEC vs. 67% for EMDA; adjusted HR 0.95; *p* = 0.4). These findings confirm that both techniques provide comparable long-term efficacy in this patient population.

When comparing our results with previous studies, it is evident that EMDA, as an adjuvant therapy in populations with similar characteristics to our cohort, offers notable benefits in recurrence reduction [8,9,10]. For instance, Zazzarra et al. [10] reported that EMDA outperforms BCG therapy in patients with NMIBC (RFS at 12 months: 72% for EMDA vs. 59% for BCG; *p* = 0.025). Di Stasi et al. [11] also demonstrated that EMDA achieves superior outcomes compared to MMC administration under normothermic conditions and results comparable to BCG in high-risk patients, further supporting its efficacy in this population. Additionally, our findings align with the recurrence-free survival (RFS) reported by Racioppi [12] in BCG-unresponsive patients (61.5% at 36 months), further validating our results.

More recent studies, such as those by Sanz [13] and Busetto [14], have highlighted the potential of EMDA combined with BCG, particularly in BCG-unresponsive patients. These studies reported complete response rates of 70% at 24 months and recurrence-free survival rates of 36% at 38 months, emphasizing the value of EMDA in specific clinical scenarios.

Regarding HIVEC, clinical trials by Angulo [15] and Tan [16] found no significant differences in recurrence-free survival rates when comparing HIVEC with MMC under normothermic conditions in intermediate-risk patients at 24 months. However, recent studies like Sachan’s [17] suggest that HIVEC may have similar efficacy to BCG in preventing recurrences. Guerrero-Ramos [18] reported comparable outcomes for HIVEC and BCG in high-risk populations at 24 months (86.5% HIVEC vs. 71.8% BCG *p* = 0.81), and the findings reported by Pazir [19] align with this trend. However, Chystiakov [20], in his cohort, even described the superiority of COMBAT after 30 months of follow-up (RFS 81.1% vs. 57.4%; *p* = 0.008).

In terms of safety, although the evaluation of adverse events was not a primary objective of our study, we observed that the adverse events associated with the evaluated treatments were mostly localized and self-limited, similar to those reported in other studies [6,9]. This differentiates them from the systemic adverse events associated with BCG therapy [4].

A significant limitation of our study is that it is not a randomized clinical trial, which could introduce biases. Nonetheless, the baseline characteristics of the patients were comparable between groups, partially mitigating this potential bias and supporting the validity of our findings. Future randomized studies could confirm our observations and provide stronger evidence. Another limitation of this study is the failure to consider prior treatments, such as BCG therapy, whose impact could be relevant to the outcomes. Additionally, while the comparison with BCG was not part of this study, this targeted approach enabled a deeper analysis of the two therapies, offering an opportunity for future studies to incorporate BCG and enrich our understanding.

Our study makes a significant contribution by providing long-term data directly comparing HIVEC and EMDA—an underexplored comparison until now. The findings reinforce the notion that both techniques are viable options with comparable outcomes in managing intermediate- and high-risk NMIBC. This has important implications for clinical decision-making, enabling healthcare professionals to select the most suitable treatment based on patient characteristics, treatment tolerability, and institutional experience.

From a broader perspective, our findings underscore the need to personalize treatments according to patient profiles, considering factors such as age, overall health status, prior response to therapies like BCG, and additional risk factors.

In this clinical landscape, new promising alternatives are emerging. For instance, the Sunrise-1 study [21], a multicenter phase IIb trial, compares TAR-200 with intravesical gemcitabine plus systemic cetrelimab against TAR-200 alone and cetrelimab alone in BCG-unresponsive high-risk NMIBC patients. Similarly, the KEYNOTE-057 [22] study has explored intravenous pembrolizumab in BCG-unresponsive patients, as well as its combination with intravesical instillations prior to BCG therapy [23]. Other alternatives include intravesical administration of gemcitabine/docetaxel [24] and, more recently, oral treatment with erdafitinib for this patient population [25].

To optimize future outcomes, studies should focus on establishing standardized treatment protocols and defining the optimal number of instillations, the ideal maintenance duration, and the most appropriate MMC dose to maximize efficacy while minimizing adverse effects. Additionally, identifying biological or clinical biomarkers predictive of response to HIVEC or EMDA could facilitate the personalization of therapeutic strategies. Exploring combinations of these devices with immunotherapies or emerging chemotherapeutic agents is also crucial, as is extending follow-up periods to evaluate the sustainability of results over the long term.

## 5. Conclusions

In conclusion, our study strengthens the evidence that both HIVEC and EMDA are effective options for managing intermediate- and high-risk NMIBC. Despite the absence of significant differences in outcomes, treatment selection should be based on an individualized assessment of each patient and the available resources. Future studies should address current uncertainties and optimize protocols to improve outcomes in this challenging disease.

## Figures and Tables

**Figure 1 cancers-17-00453-f001:**
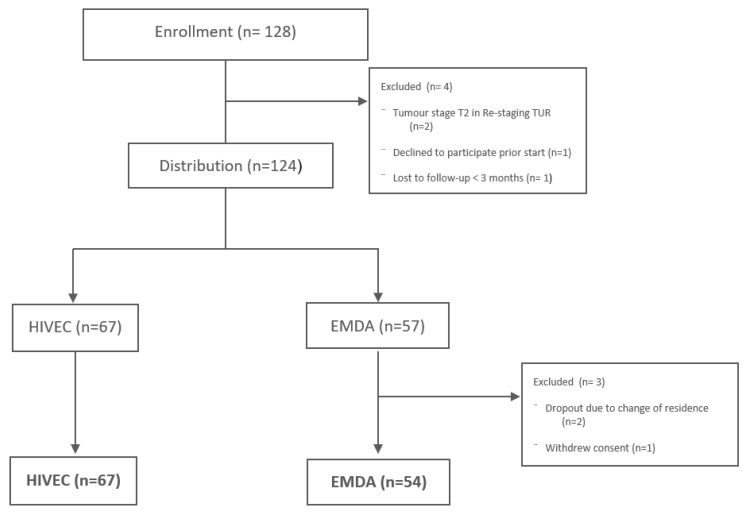
Patient flow and disposition throughout this study.

**Figure 2 cancers-17-00453-f002:**
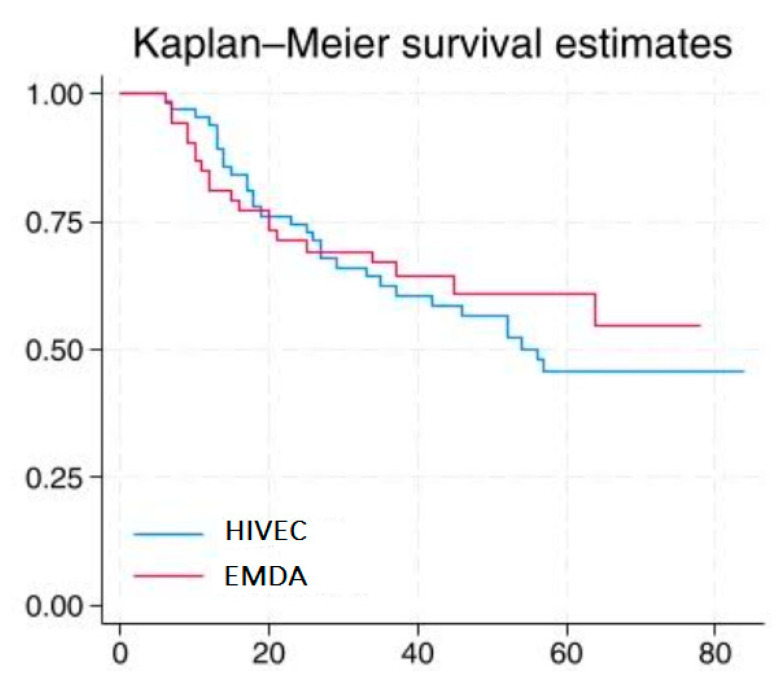
Kaplan–Meier curves for disease-free survival (DFS).

**Figure 3 cancers-17-00453-f003:**
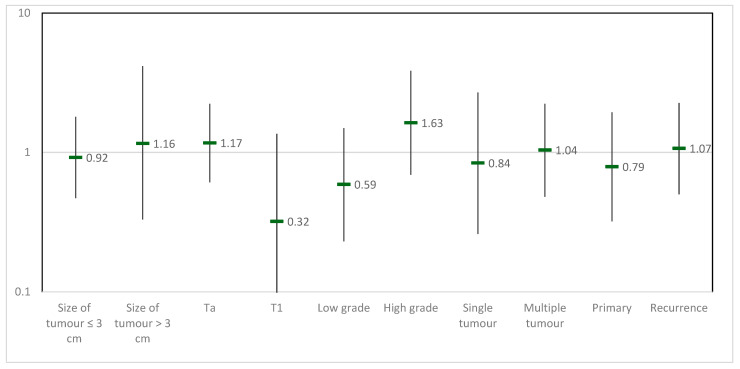
Hazard ratios for NMIBC risk factors stratified by treatment group.

**Table 1 cancers-17-00453-t001:** Description of baseline characteristics of patients according to treatment group.

		QHT 6 + 6 N = 67	EMDA 6 + 6 N = 54	*p*
Sex *n*, (%)	Men	52 (77.6%)	43 (79.6%)	*0.072*
	Women	15 (22.4%)	11 (20.4%)	
Age *n*, (%)	≤70 years	31 (46.3%)	26 (48.1%)	*0.837*
	>70 years	36 (53.7%)	28 (51.9%)	
Tumour Status *n*, (%)	Primary	38 (56.7%)	32 (56.3%)	*0.778*
	Recurrent	29 (43.3%)	22 (30.7%)	
Multiplicity *n*, (%)	No	27 (40.3%)	24 (44.4%)	*0.646*
	Múltiple	40 (59.7%)	30 (55.6%)	
Size *n*, (%)	≤3 cm	52 (77.6%)	39 (72.2%)	*0.495*
	>3 cm	15 (22.4%)	15 (27.8%)	
Stage *n*, (%)	Ta	41 (61.2%)	34 (63%)	*0.842*
	T1	26 (38.8%)	20 (37%)	
WHO Grade 2004/2016 *n*, (%)	Low	35 (52.2%)	25 (46.3%)	*0.516*
	High	32 (47.8%)	29 (53.7%)	
EAU Risk *n*, (%)	Intermediate	44 (65.7%)	32 (59.2%)	*0.468*
	High	23 (34.33%)	22 (40.7%)	

**Table 2 cancers-17-00453-t002:** Number of patients at risk for disease-free survival at each time point.

Time (Months)	10	20	30	40
Patients at risk of HIVEC	63	48	38	33
Patients at risk of EMDA	48	39	34	22

## Data Availability

The data presented in this study are available on request from the corresponding author. The data are not publicly available due to privacy issues.

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
