# Peer review of "Long-Term Outcomes of Intravesical Mitomycin C Administered via Electromotive Drug Administration or Conductive Chemo-Hyperthermia in Non-Muscle-Invasive Bladder Cancer"

_cancers, 2025, doi:10.3390/cancers17030453_

Round 1

Reviewer 1 Report

Comments and Suggestions for Authors

I suggest to improve the introduction section ( too extensive, please remove the following section “ The introduction should briefly place the study in a broad context and highlight why it is important. It should define the purpose of the work and its significance”). Any data concerning safety outcomes? Discussion: the conclusions are repeated in the discussion section. 

Author Response

Comments 1: I suggest to improve the introduction section ( too extensive, please remove the following section “ The introduction should briefly place the study in a broad context and highlight why it is important. It should define the purpose of the work and its significance”).

Response 1:Thank you for your comment, we have made modifications to the introduction to reduce its length. I hope this makes it clearer. Thank you for giving us the opportunity to improve our work

Comments 2: Any data concerning safety outcomes? 

Response 2 :Thank you for your comment. We’ve added a brief paragraph in the discussion, noting that while adverse events were not the primary focus of our study, most of the events observed were localized and self-limited, similar to those reported in other studies, distinguishing them from the systemic adverse events associated with BCG therapy.

In terms of safety, although the evaluation of adverse events was not a primary objec-tive of our study, we observed that the adverse events associated with the evaluated treat-ments were mostly localized and self-limited, similar to those reported in other studies (6, 9). This differentiates them from the systemic adverse events associated with BCG therapy (4).

Comments 3:  Discussion: the conclusions are repeated in the discussion section. 
Response 3: Thank you for this appreciation, it had gone unnoticed. It is already modified.

Reviewer 2 Report

Comments and Suggestions for Authors

Reviewer Comment:

I want to congratulate the authors to this manuscript. In this manuscript the authors conduct an observative and comparative study on two treatment alternatives for intermediate- and high-risk NMIBC. Whilst the introduction and discussion are well written, some questions remain, especially about the study design.

Abstract:

The conclusion that ‘both treatments are viable alternatives to BCG Therapy’ (Line 39) cannot be drawn from this study. This sentence should be removed. 

Introduction: 

Line 47-48 are the instructions by the journal contained in the example manuscript format and should be deleted. 

Other than that, the introduction is precise and well-constructed. 

Materials and Methods:

Line 82 Subtitle ‘Design’ is written wrong. Please correct spelling mistake. 

In the method section it is described that patients were offered standard of care (SOC). Why did the authors choose not to compare SOC with the described treatments? 

How many patients with intermediate- or high risk NMIBC were treated in total in the time-period of the conducted study? How many chose SOC over the described treatments? Please provide this data (e.g. in Flow-chart Figure 1). 

The authors should consider comparing MMC standard vs BCG vs EMDA vs HIVEC. 

Last included patient was December 2024. By the study protocol (Follow up every 3 months) this patient was not yet followed up and cannot be included in the study. Please clarify thisimportant issue. Does this concern more patients in the described cohort? The authors should include Interquartile range with the median Follow-up period of 38.6 months.

Results: 

Kaplan-Meier curve: Please show patients at risk. 

Discussion:

No improvements needed.

Conclusion: 

No improvements needed.

Author Response

Comments 1: 
Abstract:
The conclusion that ‘both treatments are viable alternatives to BCG Therapy’ (Line 39) cannot be drawn from this study. This sentence should be removed.  

Response 1: We appreciate your observation and understand your concern regarding the statement, which might indeed come across as presumptuous. Upon reflection, we recognize that our intention was to convey that these treatments hold potential as future options, based on patient characteristics and preferences, rather than asserting that they are currently viable alternatives to BCG therapy.
We have revised the text as follows:
"These findings suggest that both treatments show promise as potential future options for managing NMIBC, providing clinicians with additional considerations based on patient characteristics and preferences."

We hope this revision addresses your concern. Please do not hesitate to share any additional feedback.

Comments 2: 
Line 47-48 are the instructions by the journal contained in the example manuscript format and should be deleted. 
Other than that, the introduction is precise and well-constructed
Response 2: Thanks for noticing, it was completely a mistake. It is now corrected, thank you.

Comments 3: 
Materials and Methods:
Line 82 Subtitle ‘Design’ is written wrong. Please correct spelling mistake. 
Response 3:Correct comment, we have corrected it instantly, thank you.

Comments 4: 

In the method section it is described that patients were offered standard of care (SOC). Why did the authors choose not to compare SOC with the described treatments? 
How many patients with intermediate- or high risk NMIBC were treated in total in the time-period of the conducted study? How many chose SOC over the described treatments? Please provide this data (e.g. in Flow-chart Figure 1). 
The authors should consider comparing MMC standard vs BCG vs EMDA vs HIVEC

Response 4: We appreciate your comment and understand your concern regarding the comparison between the standard of care (SOC) and the treatments evaluated in our study.

We acknowledge that the lack of comparison with the standard treatment is a limitation of our work, as mentioned in the limitations section. However, we chose to focus exclusively on the comparison between the two described devices (HIVEC and EMDA) because no direct comparisons had been made between them previously, which provided an opportunity to assess their efficacy in a population with similar characteristics and long-term follow-up. By limiting the analysis to these two treatments, we ensured greater uniformity in treatment and follow-up protocols, which allowed us to better control variables and minimize potential biases.

Regarding the data on the total number of patients with intermediate- or high-risk NMIBC treated during the study period and how many opted for conventional treatment, we understand that this information would be useful. However, we do not have these specific data, as we only analyzed patients who chose to participate in the study, and their choice between the standard treatment and the evaluated treatments was not recorded.
"Additionally, while the comparison with BCG was not part of this study, this targeted approach enabled a deeper analysis of the two therapies, offering an opportunity for fu-ture studies to incorporate BCG and enrich our understanding."

Comments 5: 
Last included patient was December 2024. By the study protocol (Follow up every 3 months) this patient was not yet followed up and cannot be included in the study. Please clarify thisimportant issue. Does this concern more patients in the described cohort? The authors should include Interquartile range with the median Follow-up period of 38.6 months.
Response 5: We appreciate your comment and understand your concern. However, we would like to clarify that the fact the study ended in December 2024 does not mean the last patient was included on that date. According to the study protocol, patient inclusion was carried out in advance to ensure that all participants had enough time to complete at least one 3-month follow-up
Los autores deberían incluir el rango intercuartil con el período de seguimiento medio de 38,6 meses.

Thanks for your comment. We have made the change from standard deviation to interquartile range as indicated. The median follow-up period was 38.6 months (Interquartile range: 23.8-53.4 months

Comments 6: Results: 
Kaplan-Meier curve: Please show patients at risk

Response 6: We appreciate your comment. We have included a table showing the number of patients at risk at each time point of the Kaplan-Meier curve.

Reviewer 3 Report

Comments and Suggestions for Authors

Thank you for the opportunity to read and review the manuscript "Long-Term Outcomes of Intravesical Mitomycin C Administered via Electromotive Drug Administration or Conductive

Chemo-Hyperthermia in Non-Muscle Invasive Bladder Cancer" by Teresa Melgarejo-Segura et al.

A very interesting and still unresolved topic - the authors should be congratulated on their choice of research topic. Indeed, the problem of alternative therapeutic methods to BCG, especially in the group of intermediate-risk patients, is still a challenge.

The authors present long-term efficacy results comparing the two methods EMDA vs HIVEC.

The authors should be commended for their work.

The manuscript is well designed.

The language layer is correct and reader-friendly.

The discussion is balanced and specific.

Of course, the ideal comparison would be to compare both presented methods with the gold standard - BCG.

One note - since in the introduction the authors refer to BCG as generating adverse effects, the manuscript should include a paragraph devoted to the adverse effects generated by both methods tested.

Author Response

Comments 1: Of course, the ideal comparison would be to compare both presented methods with the gold standard - BCG.
One note - since in the introduction the authors refer to BCG as generating adverse effects, the manuscript should include a paragraph devoted to the adverse effects generated by both methods tested.
Response 1: Thank you for your comment. We’ve added a brief paragraph in the discussion, noting that while adverse events were not the primary focus of our study, most of the events observed were localized and self-limited, similar to those reported in other studies, distinguishing them from the systemic adverse events associated with BCG therapy.

"In terms of safety, although the evaluation of adverse events was not a primary objec-tive of our study, we observed that the adverse events associated with the evaluated treat-ments were mostly localized and self-limited, similar to those reported in other studies (6, 9). This differentiates them from the systemic adverse events associated with BCG therapy (4)."

Reviewer 4 Report

Comments and Suggestions for Authors

The authors present their prospective observational series comparing patients who receive intravesical mitomycin with HIVEC versus EDA.   Patients were not randomized to treatment modality.  Treatment modality was selected by patients following discussion with their care provider. The primary endpoint evaluated was disease recurrence.  A strength of the manuscript is 3 year follow up on some patients.

1.  Please include prior treatments received by patients (such as BCG).  This should be compared between groups and evaluated for impact on future recurrence.

2.  In patients that did have disease recurrence, was there a difference in tumor grade or stage between treatment groups?

Author Response

Comments 1: Please include prior treatments received by patients (such as BCG).  This should be compared between groups and evaluated for impact on future recurrence.

Response 1: We appreciate your valuable comment regarding the importance of including prior treatments received by patients, such as BCG, and evaluating their impact on future recurrences. We acknowledge that this is a limitation of our study, as our focus was on analyzing the EMDA and COMBAT devices, and we only considered whether patients had previously received these therapies in the inclusion and exclusion criteria.

We have included this point in the limitations section of our work:

"Another limitation of this study is the failure to consider prior treatments, such as BCG therapy, whose impact could be relevant to the outcomes."

Comments 2:  In patients that did have disease recurrence, was there a difference in tumor grade or stage between treatment groups?

Response 2: We appreciate your work and the opportunity to improve our manuscript. The information you requested is already included in the text, in the exploratory analysis section, where stratification by risk factors was performed and the impact of stage, grade, and other factors was evaluated, without finding significant differences between the treatment groups